# A Real-Time Effectiveness Evaluation Method for Remote Sensing Satellite Clusters on Moving Targets

**DOI:** 10.3390/s22082993

**Published:** 2022-04-13

**Authors:** Zhi Li, Yunfeng Dong, Peiyun Li, Hongjue Li, Yingjia Liew

**Affiliations:** School of Astronautics, Beihang University, Beijing 100191, China; lzzzzz@buaa.edu.cn (Z.L.); lipeiyun@buaa.edu.cn (P.L.); lihongjue@buaa.edu.cn (H.L.); liewyingjia@buaa.edu.cn (Y.L.)

**Keywords:** remote sensing satellite cluster, moving targets, effectiveness evaluation, simulation, neural network

## Abstract

Recently, remote sensing satellites have become increasingly important in the Earth observation field as their temporal, spatial, and spectral resolutions have improved. Subsequently, the quantitative evaluation of remote sensing satellites has received considerable attention. The quantitative evaluation method is conventionally based on simulation, but it has a speed-accuracy trade-off. In this paper, a real-time evaluation model architecture for remote sensing satellite clusters is proposed. Firstly, a multi-physical field coupling simulation model of the satellite cluster to observe moving targets is established. Aside from considering the repercussions of on-board resource constraints, it also considers the consequences of the imaging’s uncertainty effects on observation results. Secondly, a moving target observation indicator system is developed, which reflects the satellite cluster’s actual effectiveness in orbit. Meanwhile, an indicator screening method using correlation analysis is proposed to improve the independence of the indicator system. Thirdly, a neural network is designed and trained for stakeholders to realize a rapid evaluation. Different network structures and parameters are comprehensively studied to determine the optimized neural network model. Finally, based on the experiments carried out, the proposed neural network evaluation model can generate real-time, high-quality evaluation results. Hence, the validity of our proposed approach is substantiated.

## 1. Introduction

Remote sensing satellites obtain information from the Earth’s surface through optical or microwave payloads [1]. They are widely used in many fields, including agriculture, forestry, ocean, meteorology, and military [2]. With the growing emphasis on space and the ongoing growth of science and technology, the number of remote sensing satellites has expanded significantly in recent years. Meanwhile, the temporal, spatial, and spectral resolutions of satellites have also been gradually enhanced [3]. Satellite clusters, such as Planet Labs, Gaofen, and Jilin-1, are formed as a result of these advancements [4,5]. Remote sensing satellite clusters regularly consist of numerous wide-swath satellites and high-resolution satellites. They cooperate to complete census and detailed survey tasks for various targets. Nevertheless, constructing such satellite clusters is costly due to the several processes involved, including the design, manufacture, testing, launching, and management. Consequently, the effectiveness of satellite clusters has to be evaluated accurately for better decision making [6]. With the emerging drawbacks of qualitative evaluation methods, quantitative evaluation methods have received considerable attention in recent years [7,8].

The four main branches of the current quantitative effectiveness evaluation methods are the analytical method, experimental statistical method, multi-index synthesis method, and simulation method [9]. Although the analytical method is simple and efficient, it has difficulty in solving certain complex evaluation problems. Zhao [10] studied the effectiveness equivalence algorithm of the weapon system and proposed an approximate analytical expression of the weight coefficient. Next, despite the high reliability of the experimental statistical method, it requires real data, which is normally scarce and difficult to obtain. Xiong [11] used more than ten years of in-orbit data to evaluate NASA EOS Terra and Aqua MODIS on-orbit performance. The multi-index synthesis method has a good hierarchy and a wide range of applications. However, the weights of indices are usually affected by subjectiveness. Zheng [12] evaluated three typical remote sensing tasks using the analytic hierarchical process (AHP). Chen [13] constructed an evaluation indicator system to describe the target feature detection ability of satellites, where the weight parameters are obtained from the expert evaluation. Liu [14] utilized the AHP and the availability dependability capability (ADC) model to evaluate the comprehensive effectiveness of the earth observation satellite cluster. While the simulation method can consider more factors of complex systems and environments, it also requires the model to be highly accurate. Zhang [4] built a satellite observation effectiveness evaluation model based on the availability capacity profitability (ACP) framework. Tang [15] used the Satellite Tool Kit (STK) and C++ to build a nanosatellite constellation model and evaluated its military effectiveness. As one of the four branches mentioned above, the simulation method can compute the evaluation results presented in various conditions, solving the evaluation problem of the complex remote sensing satellite cluster effectiveness with good applicability [9].

The main challenge of using the simulation method to evaluate the remote sensing satellite cluster is the lack of model accuracy [9]. The satellite cluster is a complex and highly comprehensive system. Multiple physical fields, including mechanics, electricity, thermodynamics, optical, and magnetism, are coupled [16,17]. The remote sensing satellite clusters are usually used to observe three typical targets, which are point targets, regional targets, and moving targets [18]. Establishing the observation scene of the moving targets is more complex than the others, since additional aspects must be considered. Moving targets have time sensitivity and position uncertainty, requiring multi-satellite cooperation to discover, identify, confirm, and track them sequentially. The uncertainty of the observation results is mostly driven by factors, including resolution, light, cloud, and climate [18]. The traditional simulation evaluation modeling only considers a few factors, which are the satellite orbit, attitude, and optical payload visible model [4,9]. Nonetheless, apart from lacking resource constraint models, such as the power and computer storage, it also does not consider the impact of imaging quality on the mission status. Hence, to describe the complex coupling relationships of a remote sensing satellite cluster, a high-fidelity simulation model must be established [19].

Regarding the construction of indicator systems, most remote sensing satellite evaluation indicators only consider the time resolution (e.g., the revisit time and coverage time) and spatial resolution (e.g., the Ground Sampling Distance (GSD) and percentage of area coverage) [9,12]. Even though the aforementioned indicators can reflect the satellite’s availability to observe the target under ideal conditions, they are not capable of reflecting the effectiveness of the satellite’s actual mission process. Moreover, the establishment of an indicator system is generally proposed by experts, where each indicator in distinct systems might be inevitably correlated, thereby defying the principle of hierarchy and independence [20].

Furthermore, a high-fidelity simulation evaluation model requires heavy computation, especially when the cluster contains numerous satellites [21]. As a consequence, the time-consuming simulation limits the iteration speed of the design stage and the decision-making speed of the use stage [22]. A common approach to overcoming this problem is to establish a machine learning model. Machine learning technology has lately been applied to the theoretical studies of aerospace missions [23], including aircraft design [20,22], mission planning [1,24], and attitude control [25,26]. In those cases, machine learning has demonstrated its capability of learning complicated functions and responding quickly.

In this paper, by applying machine learning to the effectiveness evaluation of remote sensing satellite clusters on moving targets, a real-time evaluation model architecture is proposed. Firstly, we establish a multi-physical field coupling simulation model of the satellite cluster, which considers the repercussions of multiple satellite resource constraints and the uncertainty of imaging quality on the observation results. Secondly, we develop an indicator system to evaluate the effectiveness of satellite cluster observations on moving targets. A set of independent indicators is then filtered out through correlation analysis. Thirdly, neural networks are trained with high-fidelity simulation evaluation data. Neural networks of different hidden layers, neurons, and activation functions are trained to determine the optimized model, which can output effectiveness evaluation results in real time.

The remainder of the paper is organized as follows. In Section 2, the architecture of the real-time effectiveness evaluation model is presented. In Section 3, the high-fidelity model construction method is explained. In Section 4, the construction and screening process of the evaluation indicator system is provided. In Section 5, the neural network training method is presented. In Section 6, the experiments and results are described. In Section 7, we discuss the validity of the method proposed and conclude the paper.

## 2. Architecture of the Real-Time Satellite Cluster Effectiveness Evaluation Model

The scene of the satellite cluster observing the moving targets is illustrated in Figure 1. A satellite cluster may include different satellites, such as wide-swath satellites and high-resolution satellites. A variety of satellites in the satellite cluster collaborate to conduct the census and comprehensive investigation of the target. Wide-swath satellites are utilized to search for and discover targets in a specified area. High-resolution satellites are intended for identification, confirmation, and status tracking of discovered targets.

To evaluate the moving target’s observation effectiveness of the remote sensing satellite cluster, a real-time evaluation model architecture is designed, as shown in Figure 2.

The multi-physical field coupling simulation model of remote sensing satellite clusters is described in Section 3. The model is defined as
(1)$$Y=f_{Simulation}\left(X_{All},t\right),$$
where Y is the simulation result data and t is the simulation time. $X_{All}$ consists of simulation condition parameters, which can be expressed as
(2)$$X_{All}={\left[X_{ADCS},X_{POWER},X_{TTC\&DT},X_{PAYLOAD},X_{Space},X_{Target}\right]}^{T}.$$

The simulation condition parameters $X_{All}$ mainly consist of six parts, which are attitude determination and control subsystem (ADCS) parameters $X_{ADCS}$, the power subsystem (POWER) parameters $X_{POWER}$, the telemetry/telecommand and data transmission subsystem (TTC&DT) parameters $X_{TTC\&DT}$, the payload subsystem parameters $X_{PAYLOAD}$, the task environmental parameters $X_{Space}$, and the target parameters $X_{Target}$, respectively.

The moving target observation effectiveness indicator system of the satellite cluster is established in Section 4, which can be expressed as
(3)$$Z={\left[z_{1},\ldots z_{i},\ldots z_{n}\right]}^{T}.$$

The effectiveness evaluation model based on the simulation result data can be expressed as
(4)$$Z_{Simulation}={\left[z_{1,Simulation},\ldots z_{i,Simulation},\ldots z_{n,Simulation}\right]}^{T}=f_{SCEE-Simulation}\left(Y\right).$$

In Section 5, the neural network evaluation model is trained, which can be expressed as
(5)$$Z_{Learning}={\left[z_{1,Learning},\ldots z_{i,Learning},\ldots z_{i,Learning}\right]}^{T}=f_{SCEE-Learning}^{Stakeholders,i}\left(X_{Stakeholders,i}\right),$$
where $X_{Stakeholders,i}\in X_{All}$ contains the factors with which the ith stakeholder is concerned. $f_{SCEE-Learning}^{Stakeholders,i}$ is the corresponding neural network evaluation model. For different stakeholders, such as the designers, manufacturers, and users, $X_{Stakeholders,i}$ contains different elements. On the one hand, the primary concern of the designers and manufacturers is the composition and structure of the remote sensing satellite cluster. On the other hand, the fundamental interest of the users is to what extent different task environments and target parameters impact the overall system efficiency.

With the rapid forward propagation process of the neural network, the calculation of the satellite cluster effectiveness indicators can be greatly accelerated. The neural network model error of the ith indicator can be expressed as
(6)$$error_{i}=\left|\frac{z_{i,learning}-z_{i,simulation}}{z_{i,simulation}}\right|.$$

## 3. Establishing the Multi-Physical Field Coupling Model of the Remote Sensing Satellite Cluster

In this section, the function $f_{simulation}$ of calculating Y according to $X_{All}$ and t is established. The function is realized via a numerical simulation based on the high-fidelity model, which should be the multi-physical field coupled. The multi-physical coupling relationship is illustrated in Figure 3.

The forces and moments of celestial bodies, such as Earth, the Sun, and the Moon, are coupled with the satellites’ orbits and attitudes. The ADCS controls the satellite’s position and attitude. They have coupling relationships with the solar array attitude of the POWER, the communication link of the TTC&DT, and the range of the payload’s field of view. Sunlight and the solar array within the POWER are coupled. The remaining battery capacity of POWER is mutually coupled with the power consumed by other subsystems. TTC&DT is coupled with other subsystems by sending and receiving commands and data. The payload’s coverage capability is coupled with the satellite’s orbit and attitude. At the same time, it is also bound by resources, such as power, storage, and data transmission. The payload imaging quality is affected by factors, such as resolution, climate environment, cloudiness level, and lighting conditions, which in turn affect the target detection probability.

The typical simulation evaluation modeling only considers a few factors, which are the satellite orbit, attitude, and optical payload visible model. Nonetheless, apart from lacking resource constraint models, such as the power and computer storage, it also does not consider the impact of imaging quality on the mission status. On the basis of the existing models, this paper established three supplement models, which are (1) the power subsystem model, (2) the optical payload’s target detection probability model, and (3) the satellite cluster mission allocation model that considers the entire satellite’s resource constraints.

### 3.1. The Power Subsystem Model

The power subsystem model includes the device power consumption model, the solar array power supply model, and the battery charge and discharge model.

(1) The device power consumption model.

The total power consumption $P_{Device}$ is calculated as follows:(7)$$P_{Device}=\sum_{i=1}^{N_{Device}}S_{Device}^{i}P_{D_{i}},$$
where i is the ith device, $i\in N^{+}$, and $1\leq i\leq N_{Device}$. $N_{Device}$ is denoted as the total number of devices. The state variable $S_{Device}^{i}$ is set to 1 when the device consumes electrical energy, or else it is set to 0. $P_{Device}$ is the power that the ith device consumes.

The on–off state of the devices on board are defined as $S_{Device}=\{S_{Device}^{1},\cdots ,S_{Device}^{i},\cdots ,S_{Device}^{N}\}$, and it can be obtained by
(8)$$S_{Device}=f_{DS}\left(X_{PowerManage},X_{TC},J\left(\geq E_{0}\right)\right),$$
where $X_{TC}$ is the power management command from the Earth station, and $X_{PowerManage}$ is the management command from the on-board computer.

Denote t as the current time and Δt as the simulation time step. The power management command of t+Δt can be obtained from
(9)$$X_{PowerManage}(t+\Delta t)=f_{PM}\left(S_{Device}(t),P_{Device}(t),P_{Wing}(t),C_{Battery}(t)\right).$$

The power consumption $P_{D_{i}}$ of each device is coupled with other subsystems, and it can be expressed as
(10)$$P_{D_{i}}=f_{DP}\left(X_{D_{i}},X_{Device\_i}^{Coupling},J\left(\geq E_{0}\right)\right),$$
where $X_{D_{i}}$ is the state parameter of the ith device itself and $X_{Device\_i}^{Coupling}$ represents the state parameter affected by other subsystems, including the temperature, data transmission load, and data processing load.

(2) The solar array power-supply model.

The output power of the solar array is calculated as follows:(11)$$P_{Wing}=C_{Shadow}X_{Wing}S_{0}A_{C}\eta \cos\theta_{SW}(\beta_{P}\Delta T+1),$$
where $C_{Shadow}$ represents whether the satellite is in the Earth’s shadow, $X_{Wing}$ is the comprehensive coefficient of the solar array, $S_{0}$ is the solar constant, $A_{C}$ is the area of the solar array, η is the photoelectric conversion efficiency of the solar array, $\theta_{SW}$ is the angle between the normal direction of the solar panel and the direction of sunlight, $\beta_{P}$ is the solar array power temperature coefficient, and ΔT is the difference between the standard temperature and the current working temperature of the solar array.

$C_{Shadow}$ is calculated according to the sun vector $R_{Sun}$, the satellite position vector $R_{Sate}$, and the angle $\theta_{SS}$ between the two vectors; the formulas are as follows:(12)$$\theta_{SS}=\arccos\left(\frac{R_{Sun}\cdot R_{Sate}}{\left|R_{Sun}\right|\cdot \left|R_{Sate}\right|}\right),$$
(13)$$C_{Shadow}==\left\{ \begin{array}{cc} 1, & if\;\theta_{SS}>\frac{\pi }{2}\;and\;\left|R_{Sate}\right|\cdot \sin\theta_{SS}<R_{e}\\ 0, & else \end{array}\right..$$

$\theta_{SW}$ can be obtained from the solar vector $R_{Sun}$ and the normal vector W of the solar panel, which can be expressed as
(14)$$\theta_{SW}=\arccos\left(\frac{R_{Sun}\cdot W}{\left|R_{Sun}\right|\cdot \left|W\right|}\right).$$

The temperature difference ΔT is
(15)$$\Delta T=T_{W}-T_{O},$$
where $T_{W}$ is the current working temperature and $T_{O}$ is the standard temperature of the solar array.

(3) The battery charge and discharge model.

When the output power of the solar array exceeds the sum of total device power consumption and battery charge power, the regulator consumes the surplus power. The surplus power can be calculated by
(16)$$P_{Loss}=\left\{ \begin{array}{ccc} P_{Wing}-P_{Device}-P_{MaxCh\arg e} & , & P_{Wing}-P_{Device}>P_{MaxCh\arg e}\\ 0 & , & else \end{array}\right.,$$
where $P_{MaxCharge}$ is the maximum charging power of the battery and $P_{Loss}$ is the power consumed by the regulator.

The battery is charged only if the output power of the solar array is greater than that of the on-board devices and the current battery capacity is below its maximum value, in which case $P_{Battery}>0$. On the contrary, the battery is discharged if the electric power of the solar array is less than that of the on-board devices and the current battery capacity is above its minimum value, in which case $P_{Battery}<0$. The battery power is defined as
(17)$$P_{Battery}=\left\{ \begin{array}{l} P_{Wing}-P_{Device}-P_{Loss},C_{MinBattery}<C_{Battery}<C_{MaxBattery}\\ 0\;\;\;\;,\;\;\;\;else \end{array}\right.,$$
where $C_{MinBattery}$ is the minimum capacity of the battery and $C_{MaxBattery}$ is the maximum capacity.

The battery capacity of t+Δt can be expressed as
(18)$$C_{Battery}(t+\Delta t)=\{\begin{array}{c} C_{Battery}(t)+\Delta t\cdot k_{Charge}\cdot P_{Battery}(t),\;P_{Battery}>0\\ C_{Battery}(t)+\Delta t\cdot k_{Discharge}\cdot P_{Battery}(t),\;P_{Battery}<0 \end{array},$$
where $k_{Charge}$ is the battery charging coefficient and $k_{Discharge}$ is the battery discharge coefficient.

### 3.2. The Optical Payload Model

The optical payload model consists of two parts: the optical coverage model and the target detection probability model. The coverage state $S_{{}_{i,j}}^{Cover}$ can be obtained from the optical coverage model [9], where i is the target number and j is the task number of the ith target. The target detection probability is affected by factors, including the satellite imaging resolution, target size, light condition, cloud level, and visibility level. The detailed model is described as follows:

The resolution of the satellite optical payload is
(19)$$GSD=\frac{d}{f}\left|R_{sti}\right|,$$
where d is the pixel size, f is the focal length, and $R_{sti}$ is the vector of position difference from satellite to the target in the inertial frame.

The 2-D criterion (number cycles) N is generated by
(20)$$N=\frac{d_{t}}{GSD},$$
(21)$$d_{t}=\sqrt{L_{t}W_{t}},$$
where $d_{t}$ is the feature size of the target, $L_{t}$ is the length of the target, and $W_{t}$ is the width of the target.

The target static detection probability is defined as [27]
(22)$$P_{Stastic}=P\left(N\right)=\frac{{\left(N/N_{50}\right)}^{2.7+0.7\left(N/N_{50}\right)}}{1+{\left(N/N_{50}\right)}^{2.7+0.7\left(N/N_{50}\right)}},$$
where $N_{50}$ is the number of cycles corresponding to a 50% detection probability. The $N_{50}$ is set to 0.75, 3, 6, 1.5 for the discovery, identification, confirmation, and tracking tasks, respectively.

The solar altitude angle is calculated as
(23)$$H=\left|\arccos\left(\frac{R_{Sun}\cdot R_{Target}}{\left|R_{Sun}\right|\cdot \left|R_{Target}\right|}\right)\right|.$$

The light factor is defined as
(24)$$f_{s}=\left\{ \begin{array}{ccc} 1 & , & H\geq 30{}^{\circ}\\ 0.04\left(H-5\right) & , & 5{}^{\circ}\leq H\leq 30{}^{\circ}\\ 0 & , & H\leq 5{}^{\circ} \end{array}\right..$$

The cloud factor is generated by [28]
(25)$$f_{c}=1-\frac{{\left(N_{c}-1\right)}^{2}}{81},$$
where $N_{c}$ is the cloud level, $N_{c}\in N^{+}$, $1\leq N_{c}\leq 10$.

The visibility factor is generated by [28]
(26)$$f_{v}=e^{-\frac{{\left(N_{v}-1\right)}^{4}}{1024}},$$
where $N_{v}$ is the visibility level, $N_{v}\in N^{+}$, $1\leq N_{v}\leq 10$.

The target discovery probability can be derived from the product of static detection probability and the other three factors; it can be expressed as
(27)$$P_{i,j}=P_{Stastic}f_{s}f_{c}f_{v},$$
where i is the target number and j is the task number of the ith target.

According to the target detection probability, it is possible to succeed and fail in a single observation mission. Thus, the simulation results may be different, given the same simulation condition. The task status is defined as follows:(28)$$S_{i,j}^{Task}=\left\{ \begin{array}{cc} 1, & if\;S_{{}_{i,j}}^{Cover}=1\;and\;P_{i,j}\geq random(0,1)\\ 0, & else \end{array}\right..$$

Finally, the status and the end time of each task are recorded, including the following items:

(1) The time series of discovery tasks is recorded as $T_{i,j}^{Discovery}$, where $1\leq j\leq N_{i}^{Discovery}$. $N_{i}^{Discovery}$ is the total discovery task number of the ith target.

(2) The time series of identification and confirmation tasks is recorded as $T_{i,j}^{Identification}$, where $1\leq j\leq N_{i}^{Identification}$. $N_{i}^{Identification}$ is the total confirmation task number of the ith target.

(3) The time series of successful tracking tasks is recorded as $\left[T_{i,j}^{TrackingStart},T_{i,j}^{TrackingEnd}\right]$, where $1\leq j\leq N_{i}^{Tracking}$. $N_{i}^{Tracking}$ is the total tracking task number of the ith target.

(4) The time series of losing the targets is recorded as $T_{i,k}^{Lost}$, where $1\leq k\leq N_{i}^{Lost}$. $N_{i}^{Lost}$ is the total lost counts of the ith target. The $T_{i,k}^{Lost}$ is recorded when the simulation starts or the tasks of the target i fail twice in a row.

### 3.3. The Satellite Cluster Mission Allocation Model

The moving target observation mission can be divided into four tasks, which are discovery, identification, confirmation, and tracking. The discovery task refers to the scanning and discovery of regional targets when the potential area of the target is known, but the specific location is not. The identification and confirmation tasks refer to the identification of the target type and model when the approximate target location has been found. The tracking task refers to the continuous tracking of the target when the specific model of the target has been confirmed. The discovery task is generally completed by satellites with a low resolution and a wide field of view. As for the identification, confirmation, and tracking tasks, they are typically accomplished by satellites with a high resolution but a narrow field of view.

(1) High-resolution satellites mission allocation model.

The mission allocation steps for the identification, confirmation, and tracking tasks are designed as Figure 4, where i is the target index and $1\leq i\leq N_{Target}$, $N_{Target}$ is the total number of the targets. j is the satellite index and $1\leq j\leq N_{Sate}$, $N_{Sate}$ is the total number of satellites.

Step 1: obtain the planning current time t and planning period $\Delta t_{Plan}$, then catalog the discovered targets.
(29)$$Target=\left\{Target_{1},\ldots Target_{i},\ldots Target_{n}\right\},$$
where $Target_{i}$ represents the set of the ith target’s state information, which comprises the target index; the latest observation state; the latest observation time; the latest identified type and model; and the latest positioning longitude, latitude, speed, and course angle.

Step 2: within a planning cycle $\Delta t_{Plan}$, the positions of the targets are determined by their latest observed position and speed.

Step 3: the observation time window of all satellites for all targets during the planning period $\Delta t_{Plan}$ is calculated. The observation time window of satellite j on target i is defined as [$T_{i,j}^{WindowStart}$, $T_{i,j}^{WindowEnd}$] [1].

Step 4: the mission profit for each target is presented as
(30)$$Profit_{i}=\left\{ \begin{array}{ll} 1, & DiscoveryState\\ 0.8, & IdentificationState\\ 0.5, & ConfirmState\\ k\left(t-t_{i}\right), & TrackingState \end{array}\right.,$$
where t is the current time, $t_{i}$ is the last observed time of the ith target, and k is the time coefficient. The value of k is set to 0.001.

Step 5: the total observation cost of each satellite to the observable target is calculated, including the time cost, power consumption cost, and storage consumption cost.

The time cost is
(31)$$c_{t_{i,j}}=T_{i,j}^{WindowStart}-t.$$

The power consumption cost is
(32)$$c_{p_{i,j}}=\left(P_{Imaging}+k\cdot P_{DT}\right)\cdot \left(T_{i,j}^{WindowEnd}-T_{i,j}^{WindowStart}\right)+P_{Maneuver}\cdot \left(T_{i,j}^{ManeuverEnd}-T_{i,j}^{ManeuverStart}\right),$$
where $P_{Imaging}$, $P_{DT}$, $P_{Maneuver}$ are the average powers of the whole satellite during the imaging process, the data transmission process, and the attitude maneuver process, respectively. k is the comprehensive coefficient denoting the ratio of remote sensing payload data rate to the data transmission rate. $T_{i,j}^{ManeuverStart}$ and $T_{i,j}^{ManeuverEnd}$ are the start and end times of the attitude maneuver, respectively, which can be achieved according to the start time of the time window, the expected angle of the attitude maneuver, the maximum angular velocity and angular acceleration of the satellite attitude maneuver, and the stabilization time of the attitude maneuver [1].

The storage consumption cost is defined as
(33)$$c_{m_{i,j}}=MEM_{j}\cdot \left(T_{i,j}^{WindowEnd}-T_{i,j}^{WindowStart}\right),$$
where $MEM_{j}$ is the remote sensing payload data rate.

The total observation cost is the product of the above three costs, which is expressed as
(34)$$c_{i,j}=c_{t_{i,j}}\cdot c_{p_{i,j}}\cdot c_{m_{i,j}}.$$

Step 6: allocate the missions to the satellites. The most profitable mission among the remaining missions is chosen and allocated to the satellite that has the lowest cost to fulfill that mission while satisfying the constraints of attitude maneuver, fixed storage, or power supply. The mission’s allocation process loops until all the missions are completed. 

(2) Wide-swath satellites mission allocation model.

The allocation process of the discovery stage is similar to that of the identification, confirmation, and tracking stages mentioned above, with two main differences.

The first difference is the target cataloging. As shown in Figure 5, the potential emergence area of the target is meshed by a 1° × 1° grid of longitude and latitude, and the center of each grid is regarded as a “target”. The square boundaries of the potential area are determined by the following method. The target is located at the left boundary of the potential area at time $t_{0}$. Meanwhile, the right boundary of the potential area can be calculated according to the maximum speed of 30 knots (15.43 m/s) away from the area’s left boundary.

The second difference is the observation profit of the grid target. The profit is calculated as follows:(35)$$Profit=t-t_{i},$$
where t is the current time and $t_{i}$ is the last observed time of the ith grid.

## 4. The Effectiveness Evaluation Indicator System

In this section, we propose an evaluation indicator system of moving target observation performance to judge the effectiveness of satellite cluster observation tasks. The evaluation indicator system is screened through correlation analysis to form an independent indicator set. The following describes the methods for developing and filtering the effectiveness evaluation indicator system.

### 4.1. The Construction of the Evaluation Indicator System

The moving target observation effectiveness indicator system considers three abilities: search and discovery, identification and confirmation, and continuous tracking. The details of the effectiveness evaluation indicator system are shown in Figure 6.

The concept definition and mathematical description of the effectiveness indicators are as follows:

Denote i as the target index and $1\leq i\leq N_{Target}$, $N_{Target}$ as the total number of targets, $T_{Total}$ as the total simulation time, and $T_{Orbit}$ as the orbit period of the satellite cluster.

The search and discovery ability.

The target discovery probability $P_{Discovery}$ is defined as the average probability that the targets are rediscovered within a single orbit period from the moment they are lost. The discovery response time $T_{Discovery}$ is defined as the average time taken to rediscover a lost target. The calculation procedures are as follows:

Search and obtain the time $T_{i,j}^{Discovery}$ immediately after $T_{i,k}^{Lost}$. The time combination is recorded as $\left[T_{i,k}^{Lost},T_{i,k}^{Discovery}\right]$. The total number of discovery tasks $N_{Discovery}$ found within a single orbit period is generated by
(36)$$N_{Discovery}=\sum_{i=1}^{N_{target}}N_{i}^{Lost},\;T_{i,k}^{Discovery}-T_{i,k}^{Lost}\leq T_{Orbit},$$
where k is the kth loss of the ith target. $N_{i}^{Lost}$ is the total lost number of the ith target and $1\leq k\leq N_{i}^{Lost}$.

$P_{Discovery}$ is computed by
(37)$$P_{Discovery}=\frac{N_{Discovrey}}{\sum_{i=1}^{N_{Target}}N_{i}^{Lost}}.$$

$T_{Discovery}$ is computed by
(38)$$T_{Discovery}=\frac{\sum_{i=1}^{N_{Target}}\sum_{k=1}^{N_{i}^{Lost}}\left(T_{i,k}^{Discovery}-T_{i,k}^{Lost}\right)}{\sum_{i=1}^{N_{Target}}N_{i}^{Lost}}.$$

2.The identification and confirmation ability.

The target identification probability $P_{Identification}$ is defined as the average probability of identifying and confirming targets within a single orbit period since targets are discovered. The identification response time $T_{Identification}$ is defined as the average time taken from target discovery to identification. The calculation procedures are as follows:

Search and obtain the time $T_{i,j}^{Identification}$ immediately after the time $T_{i,j}^{Discovery}$. The time combination is recorded as $\left[T_{i,j}^{Discovery},T_{i,j}^{Identification}\right]$. The total number of successful confirmation tasks within a single orbit period since targets are detected is produced by
(39)$$N_{Identification}=\sum_{i=1}^{N_{Target}}N_{i}^{Discovery},\;T_{i,j}^{Identification}-T_{i,j}^{Discovery}\leq T_{Orbit},$$
where j means the jth discovery of the ith target. $N_{i}^{Discovery}$ is the total discovery number of the ith target and $1\leq j\leq N_{i}^{Discovery}$.

$P_{Identification}$ is computed by
(40)$$P_{Identification}=\frac{N_{Identification}}{\sum_{i=1}^{N_{Target}}N_{i}^{Discovery}}.$$

$T_{Identification}$ is computed by
(41)$$T_{Identification}=\frac{\sum_{i=1}^{N_{Target}}\sum_{k=1}^{N_{i}^{Detection}}\left(T_{i,j}^{Identification}-T_{i,j}^{Discovery}\right)}{\sum_{i=1}^{N_{target}}N_{i}^{Discovery}}.$$

3.The continuous tracking ability.

The tracking time percentage $A_{Tracking}$ is defined as the average ratio of total target tracking time to the total runtime.
(42)$$A_{Tracking}=\frac{\sum_{i=1}^{N_{Target}}\sum_{j=1}^{N_{i}^{Tracking}}\left(T_{i,j}^{TrackingEnd}-T_{i,j}^{TrackingStart}\right)}{N_{Target}\cdot T_{Total}},$$
where j means the jth tracking tasks of the ith target. $N_{i}^{Tracking}$ is the total tracking number of the ith target and $1\leq j\leq N_{i}^{Tracking}$.

The average tracking interval $T_{Tracking}$ is defined as the average interval between two consecutive tracking tasks.
(43)$$T_{Tracking}=\frac{\sum_{i=1}^{N_{Target}}\sum_{j=1}^{N_{i}^{Tracking}-1}\left(T_{i,j+1}^{TrackingStart}-T_{i,j}^{TrackingEnd}\right)}{\sum_{i=1}^{N_{Target}}N_{i}^{Tracking}}.$$

### 4.2. Evaluation Indicator Screening

The correlation of the indicator system $Z={\left[z_{1},\ldots ,z_{n}\right]}^{T}$ is analyzed, and the indicators with strong correlations are then screened and eliminated. This results in an indicator system that satisfies the principles of hierarchy and independence. The steps for evaluation indicator screening are as follows:

Step 1: calculate the correlation coefficients between the parameters and build the coefficient matrix as
(44)$$A={\left[a_{ij}\right]}_{n\times n}=\left[\begin{array}{cccc} a_{11} & a_{12} & \ldots & a_{1n}\\ a_{21} & a_{22} & \ldots & a_{2n}\\ \ldots & & & \\ a_{n1} & a_{n2} & \ldots & a_{nn} \end{array}\right],$$
where $a_{ij}$ is the correlation coefficient of the indicators i and j, note that the value of the main diagonal is 1.

Step 2: screen out the highly correlated indicators. If $a_{ij}>0.5$, then indicators i and j are considered as highly correlated.

Step 3: calculate the sum of the linear correlation coefficients of indicators i and j with the other indicators.
(45)$$C_{i}=\sum_{k=1}^{n}\left|a_{ik}\right|k\neq i\;\mathrm{and}\;k\neq j,$$
(46)$$C_{j}=\sum_{k=1}^{n}\left|a_{jk}\right|k\neq i\;\mathrm{and}\;k\neq j.$$

Step 4: compare the values of $C_{i}$ and $C_{j}$. Remove the indicator with the larger value and keep the indicator with the smaller value.

Step 5: continue Steps 2 to 4, until the indicator system is screened and formed.

## 5. Neural Network Evaluation Model Training

The refining of simulation model granularity enhances the simulation accuracy, but diminishes the simulation efficiency. Hence, it is challenging to meet the efficiency requirement of both the iterative optimization at the designer end and the real-time decision making at the user end. To provide a solution to this problem, a backpropagation (BP) neural network model is designed and trained for user stakeholders to realize the rapid evaluation of satellite cluster effectiveness. This section introduces sample generation and neural network training in detail.

### 5.1. Sample Creation

Each sample has an input and an output. The sample input is defined as
(47)$$X={\left[t_{0},Long,Lat,N_{c},N_{v},N_{Target}\right]}^{T},$$
where $t_{0}$ is the task start time, Long and Lat are the longitude and latitude for the center of the initial area, $N_{c}$ is the cloud level, $N_{v}$ is the visibility level, and $N_{Target}$ is the total target number.

The sample output is the effectiveness indicator, which can be expressed as follows
(48)$$Z={\left[z_{1},\ldots ,z_{n}\right]}^{T}.$$

A single-time simulation has high uncertainty resulting from the uncertainty of the random target initial parameters and the existence of the detection probability. Therefore, in order to generate reliable samples, each simulation case is performed multiple times to obtain the statistical value of the effectiveness indicators.

Denote $N_{Sample}$ as the total sample number and $N_{Condition}$ as the designated simulation times of a single sample. The complete sample set $S={\left[S_{1},\ldots ,S_{n}\right]}^{T}$ is generated after $N_{Sample}\times N_{Condition}$ times of simulation. $S_{i}$ is the matrix corresponding to the ith effectiveness indicator of the sample set, which has $N_{Sample}$ rows and seven columns.

### 5.2. Network Training

The neural network training process is divided into three different stages, which are the sample set division, network training, and performance testing, as shown in Figure 7.

To begin, the sample sets $S_{i}$ are first normalized to [−1,1] and then split into a training set $S_{i}^{Train}$ and a test set $S_{i}^{Test}$. Denote σ∈[0,1] as the fraction of the sample training set. $S_{i}^{Train}$ has $\sigma \cdot N_{Sample}$ rows whilst $S_{i}^{Test}$ has $\left(1-\sigma \right)\cdot N_{Sample}$ rows.

Next, deep neural networks are trained in two steps, starting with the search for the best activation function, followed by the selection of the optimal network parameters. In the first step, a single hidden layer neural network traverses a set of activation functions to find several optimal activation functions. Those activation functions are softmax, tansig, logsig, elliotsig, poslin, purelin, radbas, satlin, satlins, and tribas. In the second step, a multi-hidden-layer neural network is implemented to determine the best combination of network structures and parameters. The neural network traverses the various combinations of network structures and parameters, including the number of hidden layers, the number of neurons in each layer, and the activation functions found in the first step.

Finally, the neural networks are tested on the training set and test set. Denote $Z_{i}^{Train}$ and $Z_{i}^{Test}$ as the outputs of the neural network in the training and test set, respectively. The mean squared error (MSE) of the ith effectiveness indicator can be expressed as
(49)$$MSE_{i}^{Train}=\frac{1}{\sigma \cdot N_{Sample}}\cdot \sum_{j=1}^{\sigma \cdot N_{Sample}}{\left(Z_{i,j}^{Train}-{\hat{Z}}_{i,j}^{Train}\right)}^{2}.$$

The average error of the training and test sets is defined as
(50)$${\bar{e}}_{i}^{Train}=\frac{1}{\sigma \cdot N_{Sample}}\cdot \sum_{j=1}^{\sigma \cdot N_{Sample}}error_{i,j}^{Train},$$
(51)$${\bar{e}}_{i}^{Test}=\frac{1}{\left(1-\sigma \right)\cdot N_{Sample}}\cdot \sum_{j=1}^{\left(1-\sigma \right)\cdot N_{Sample}}error_{i,j}^{Test}.$$

## 6. Experiments and Results

In this section, the ship target observation scene is selected for the experiments. Firstly, we compared the proposed model with the model mentioned in the reference [6,9]. This is to investigate the influence of the high-fidelity model on the effectiveness evaluation results. Secondly, we used the proposed evaluation indicator system to calculate the effectiveness of the satellite cluster. Additionally, we employed the correlation analysis to filter the evaluation indicator system. The resulting indicator set was highly hierarchical and independent. Finally, we trained neural networks for effectiveness evaluation. To find the optimal network structure and parameters, numerous combinations of them traversed the neural network. As a result, the neural networks can output the effectiveness indicators instantaneously.

### 6.1. The Comparison of Different Simulation Model Granularities

The remote sensing satellite cluster is designed according to the Walker Constellation of solar synchronous orbit, which is composed of wide-swath satellites and high-resolution satellites. Each orbital plane of the cluster contains ten satellites with an interval phase of 36°. The wide-swath satellites lie in the first and sixth positions of each orbital plane, whereas the high-resolution satellites lie in the other eight positions. The satellite cluster configuration is shown in Figure 8.

The key parameters of the satellite cluster are shown in Table 1.

The mission area for this experiment is restricted to the Pacific, from 130° E 10° N to 150° E 30° N. The ships are initialized within a random region of 2° × 2°. The positions, velocities, and course angles are also arbitrarily selected. The ship size used here is 155.3 m × 20.4 m. The simulation starts at 0:00 a.m. on one day in 2021 and lasts for 21,600 s. The randomly generated environmental parameters used for the model comparison are shown in Table 2.

The initial information of ten arbitrary ships is presented in Table 3.

We built three simulation models of different granularity according to refs. [6,9], and our granularity, respectively, in which each is labeled as granularity 1, granularity 2, and granularity 3. Their model elements are presented in Table 4. Granularity 1 only considers the satellite’s orbit, attitude, and optical payload’s coverage model. On top of granularity 1, granularity 2 not only includes a data transmission model, but also considers the impact of weather uncertainty. As compared with the first two granularities, additionally, granularity 3 takes into consideration the constraints of the satellite power subsystem, as well as the influence of resolution, climatic conditions, and cloudiness level on the imaging detection probability.

The simulation process of satellite cluster observing the moving target is shown in Figure 9. The wide-swath satellites and the high-resolution satellites completed the task of discovering, identifying, confirming, and tracking the ship in turn.

The calculated effectiveness evaluation indicators of the three model granularities are presented in Table 5 and Figure 10a. The effectiveness values for granularity 1 and granularity 2 are higher than the proposed granularity 3. By comparing granularity 1 to granularity 3, it can be observed that the ${\bar{T}}_{Identification}$ and ${\bar{T}}_{Tracking}$ errors of granularity 1 are above 75%, and its error of ${\bar{A}}_{Tracking}$ is 7 times larger. As for granularity 2, even though its error, when compared to granularity 1, is lower than that of granularity 3, it still contains a large error, where the error of ${\bar{T}}_{Identification}$ is 37% and ${\bar{A}}_{Tracking}$ is 64%. The main reason behind the errors is that the coarse-grained model ignores the power constraint and lacks the imaging detection probability model, causing the model to differ significantly from the true model. Therefore, the calculated indicators are more impractical and falsely higher, which cannot be achieved by the actual satellite during operation. The limitations of coarse-grained models and the necessity of fine-grained models in effectiveness evaluation problems are proven.

The improvement of the model fidelity leads to a decrement in computational efficiency. The simulations of the three model granularities were executed on a computer with Windows 10 and Intel i7-9700 @ 3.0 GHz CPU. The time taken to perform a single simulation under the three granularities was recorded, as shown in Figure 10b. From the statistical data, it is noticeable that the time consumed by granularity 3 is 1.67 times longer than granularity 1 and 1.21 times longer than granularity 2.

### 6.2. Sample Creation

A large quantity of sample data is required for both screening the indicator system and training the neural network. In accordance with the stakeholders’ requirements, data points are randomly scattered taking the mission and environmental parameters as the sample input. The range of the sample input task parameters is shown in Table 6.

We randomly generated 1000 samples as the inputs. For each sample, a set of 20 random simulation parameter combinations were created. The parameters included the position, velocity between 0 and 10 m/s, and course angle between 0° and 360°. After 1000 × 20 = 20,000 simulations, 1000 samples were produced. Subsequently, the effectiveness indicator calculations were carried out on the samples. The first ten samples are shown in Table 7.

### 6.3. Evaluation Indicator Screening

The initial evaluation indicator system of the moving target observation is
(52)$$Z={\left[z_{1},\ldots ,z_{n}\right]}^{T}=[{\bar{P}}_{Discovery},{\bar{T}}_{Discovery},{\bar{P}}_{Identification},{\bar{T}}_{Identification},{\bar{A}}_{Tracking},{\bar{T}}_{Tracking}{]}^{T}.$$

We computed the correlation coefficients between the indicators. The correlation matrix A is presented below.
$$A=\left[\begin{array}{cccccc} 1 & -0.8059 & 0.3061 & 0.0346 & 0.3439 & 0.2159\\ -0.8059 & 1 & -0.2010 & -0.2326 & -0.3792 & -0.4441\\ 0.3061 & -0.2010 & 1 & -0.6342 & -0.1377 & -0.0857\\ 0.0346 & -0.2326 & -0.6342 & 1 & -0.3775 & 0.2925\\ 0.3439 & -0.3792 & -0.1377 & -0.3775 & 1 & -0.3685\\ 0.2159 & -0.4441 & -0.0857 & 0.2925 & -0.3685 & 1 \end{array}\right]$$

The correlation coefficient of the discovery probability ${\bar{P}}_{Discovery}$ and discovery response time ${\bar{T}}_{Discovery}$ is −0.8059, indicating a strong linear correlation between these two indicators. The sum of the correlation coefficient between ${\bar{P}}_{Discovery}$ with the other indicators is
$$C_{1}=0.3061+0.0346+0.3439+0.2159=0.9005$$

The sum of the correlation coefficient between ${\bar{T}}_{Discovery}$ with the other indicators is
$$C_{2}=0.2010+0.2326+0.3792+0.4441=1.2569$$

From a comparison of $C_{1}$ and $C_{2}$, the discovery response time ${\bar{T}}_{Discovery}$, which has a higher value, is removed.

Likewise, the correlation coefficient of the identification probability ${\bar{P}}_{Identification}$ and identification response time ${\bar{T}}_{Identification}$ is 0.6342, which means that these 2 indicators are strongly correlated. The sum of the correlation coefficient between ${\bar{P}}_{Identification}$ with the other indicators is
$$C_{3}=0.3061+0.1377+0.0857=0.5295$$

The sum of the correlation coefficient between ${\bar{T}}_{Identification}$ with the other indicators is
$$C_{4}=0.0346+0.3775+0.2925=0.7046$$

From a comparison of $C_{3}$ and $C_{4}$, the identification response time ${\bar{T}}_{Discovery}$ is deleted.

As a result, the evaluation indicator system contains four indicators, which are the discovery probability ${\bar{P}}_{Discovery}$, identification probability ${\bar{P}}_{Identification}$, tracking time percentage ${\bar{A}}_{Tracking}$, and average tracking interval ${\bar{T}}_{Tracking}$.

### 6.4. Neural Network Training

A total of 1000 samples were randomly shuffled and divided into a training set of 800 samples and a test set of 200 samples. A single hidden layer network structure was used in the traversal of activation functions mentioned in Section 5.2. Additionally, the neuron number of the layer ranged from 20 to 100. After training, the ten optimal networks are shown in Table 8.

The five best activation functions were found, which are softmax, poslin, satlin, tansig, and logsig. Then, the multi-layer network structure and parameters were tested, including the number of hidden layers (2/3/4), the number of neurons in each layer (20–200), and the five best activation functions. The best neural network of each effectiveness indicator is summarized in Table 9.

The training and test results are illustrated in Figure 11, Figure 12, Figure 13 and Figure 14. The first panel demonstrates the training convergence process of the best neural network. It is obvious that only the neural network for discovery probability has an MSE value of above 0.02, while the MSE values for the other three neural networks are all below 0.01, demonstrating the efficacy of neural network training. The second panel shows the evaluation accuracy of the ten optimal neural networks compared to the simulation evaluation samples. The third panel indicates the performance of the networks on the test set. The results show that the majority of the errors are below 10% and the maximum error is below 20%. Therefore, the validity and generalization ability of the proposed neural network model are verified.

The time consumption of the effectiveness evaluation generated by the simulation method and the neural network model is shown in Figure 15. The average time consumption of a single simulation sample obtained through 20 simulations is 0.404 × 20 = 8.074 h. By comparison, the network can output the evaluation indicators in real time.

## 7. Discussion and Conclusions

Remote sensing satellite clusters are usually comprised of wide-swath satellites and high-resolution satellites. They are playing an increasingly important role in the remote sensing field owing to their ability to complete census and detailed survey tasks for various targets. Due to the high cost of development and operation, remote sensing satellite clusters require quantitative effectiveness evaluation to support decision making in their entire life cycle. The effectiveness evaluation is usually based on simulation, but there is a conflict between accuracy and speed. Thus, the main goal of this paper was to present an architecture to achieve real-time high-quality effectiveness evaluation of remote sensing satellite clusters. The significant advantages of the architecture are as follows:The simulation model in the architecture is a multi-physical field coupled. Apart from considering the repercussion of on-board resource constraints, it also considers the consequence of imaging’s uncertainty on the observation results. As compared with our proposed model granularity, the traditional coarse-grained model has a maximum error of more than 60%, which proves the effectiveness of the proposed model’s granularity.A moving target observation effectiveness indicator system of the satellite clusters is established. In comparison with the current indicator system that contains accessibility indicators, such as coverage and resolutions, our proposed indicators can better reflect the effectiveness of the operation process. Moreover, we screened the indicator system through model-based quantitative analysis. This not only reduces the redundancy of the indicator system developed by the domain experts, but also increases the indicator system’s hierarchy and independence.The neural network model can be trained in the architecture to evaluate the effectiveness with real-time computation. The architecture supports the finding of the best network structure and parameters, including the number of hidden layers, as well as the number of neurons and activation function in each layer. The result indicates that the neural network model not only achieves high accuracy on the training set, but also generalizes well on the test set. The conflict between the accuracy and speed is therefore resolved.

Our suggested method is not merely applicable to the effectiveness evaluation of remote sensing satellite clusters, but also single satellites and other types of satellites. Yet, our method has no obvious advantage over the coarse-grained model when only low accuracy is required. A coarse-grained model is more advantageous as it already has high computational efficiency. On the contrary, the sample creation method that we mentioned demands high computational resources.

Future research will be undertaken in two aspects, which are increasing the sample size and sample quality. This could be useful in reducing the computational resources needed in neural network training and enhancing the accuracy of the neural network. Regarding the sample size issue, due to the limitation of computing resources, only 1000 samples are created to be applied in our network training process. The error of networks might be reduced if more training samples are generated. In the future, the simulation model can be further optimized to acquire as many samples as possible with the same computational resources. For example, the relationship between effectiveness and model granularity can be thoroughly investigated, so as to refine the granularity of the important parts and reduce the granularity of the less important ones. In terms of sample quality, the noise in the samples will cause errors in the network evaluation results. It is a worthwhile research direction in the future to improve the training effect by preprocessing the samples.

## Figures and Tables

**Figure 1 sensors-22-02993-f001:**
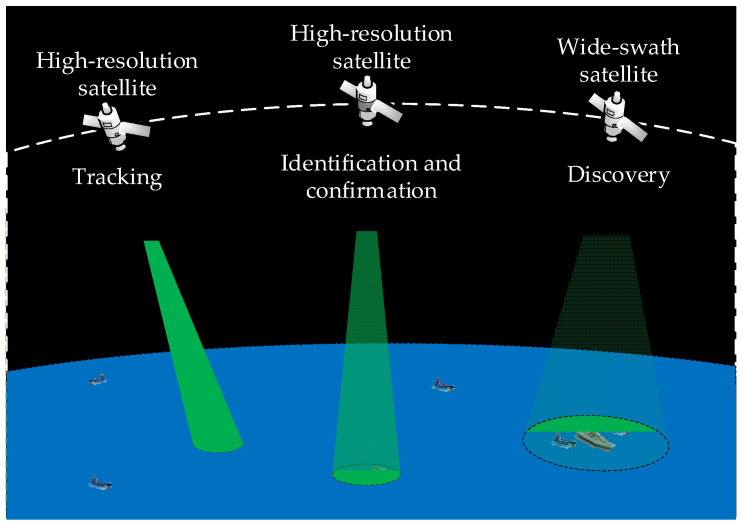
The scene of the satellite cluster observing the moving targets.

**Figure 2 sensors-22-02993-f002:**
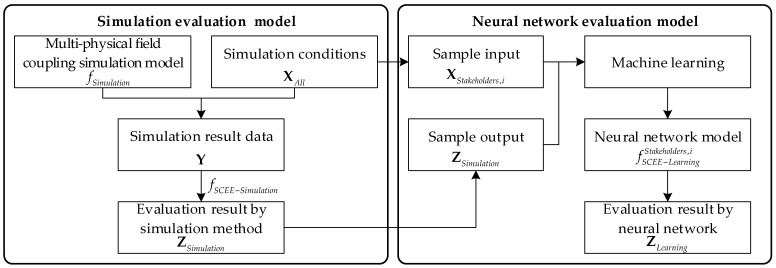
The real-time satellite cluster effectiveness evaluation model architecture.

**Figure 3 sensors-22-02993-f003:**
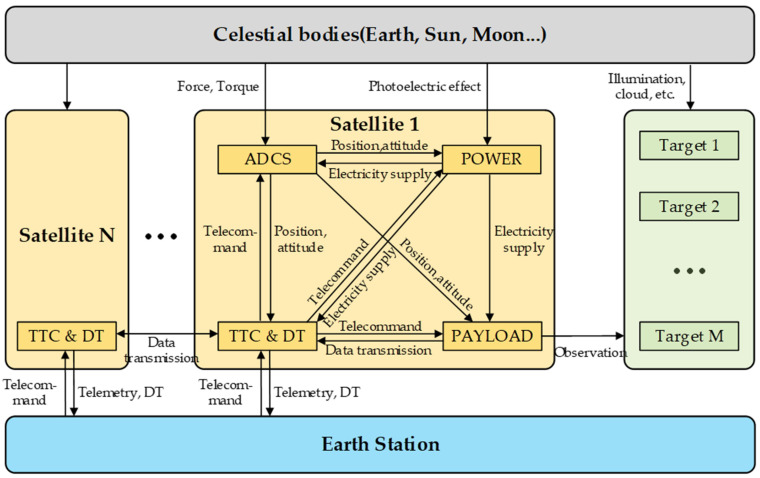
Multi-physical field coupling model of the satellite cluster.

**Figure 4 sensors-22-02993-f004:**
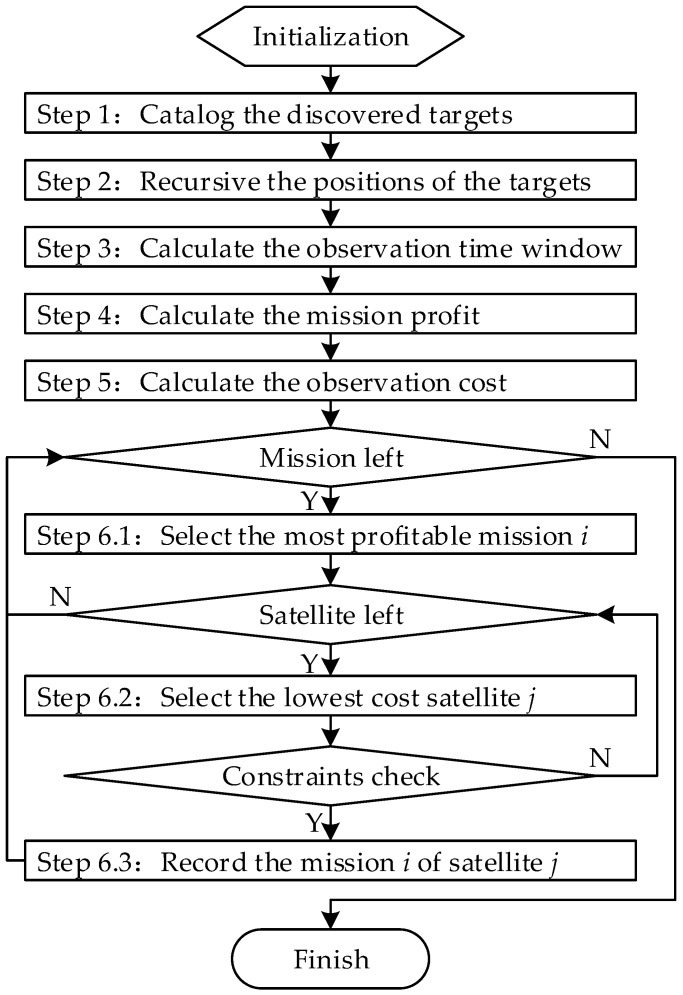
High-resolution satellites mission allocation model.

**Figure 5 sensors-22-02993-f005:**
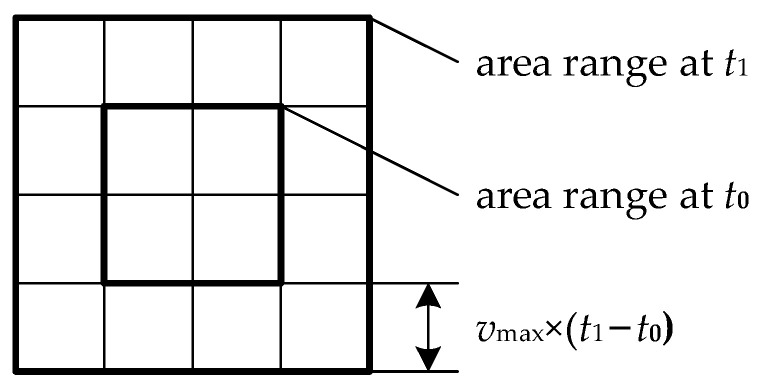
Regional grid target of the discovery task.

**Figure 6 sensors-22-02993-f006:**
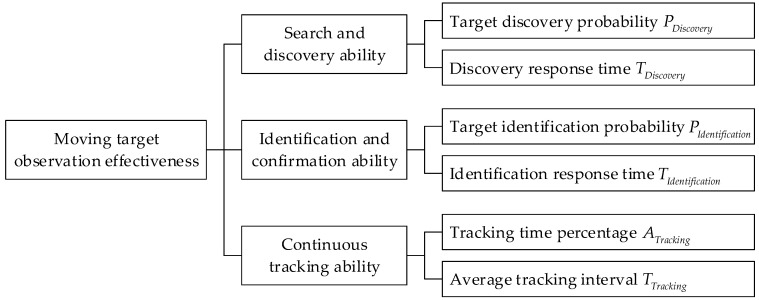
Effectiveness evaluation indicator system.

**Figure 7 sensors-22-02993-f007:**
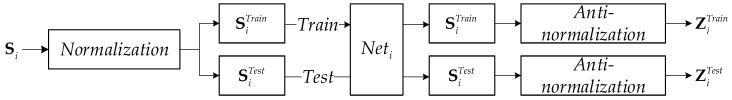
The training process of the neural network.

**Figure 8 sensors-22-02993-f008:**
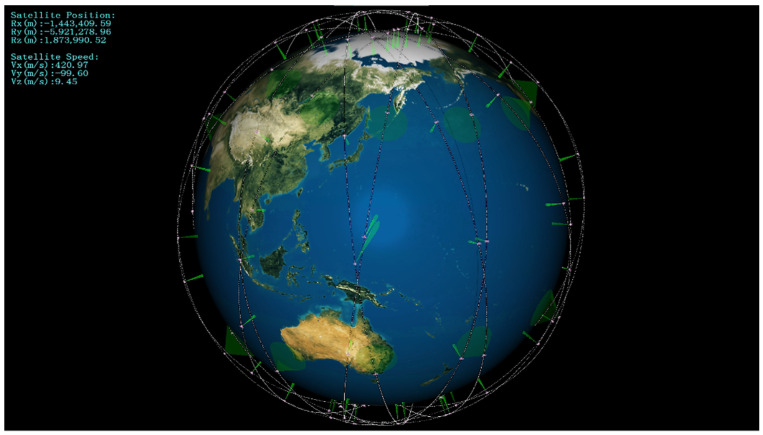
The satellite cluster configuration.

**Figure 9 sensors-22-02993-f009:**
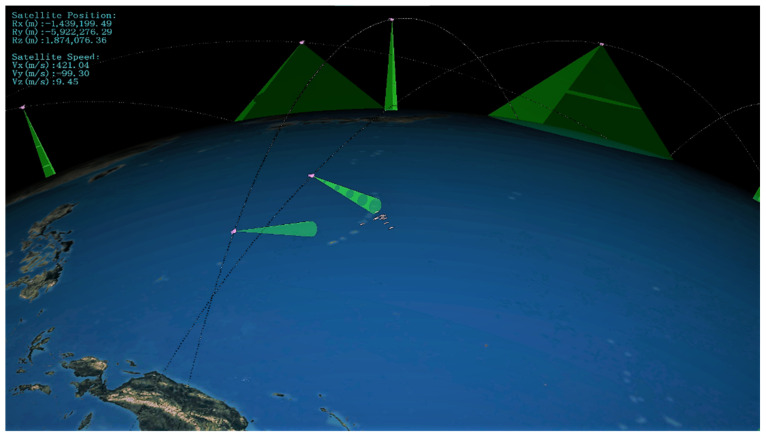
Moving targets’ observation missions.

**Figure 10 sensors-22-02993-f010:**
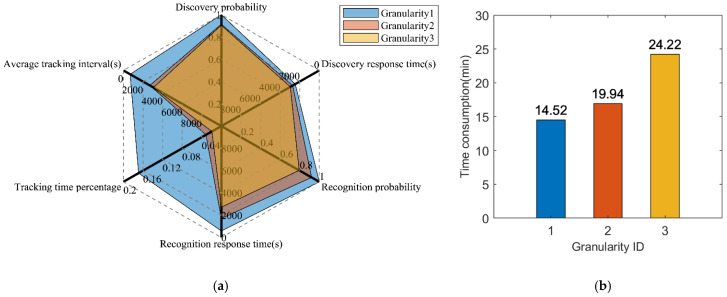
The comparison results of the effectiveness evaluation and time consumption for different simulation model granularities. (**a**) Evaluation results comparison of the three different simulation model granularities. (**b**) Time consumption comparison of the three different simulation model granularities.

**Figure 11 sensors-22-02993-f011:**
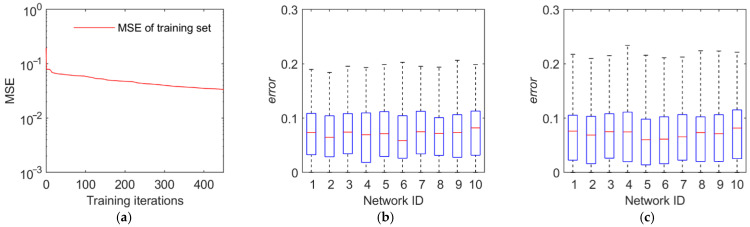
Training and test results of the discovery probability. (**a**) Training convergence process of the best neural network. (**b**) Errors of the ten best networks on the training set. (**c**) Errors of the ten best networks on the test set.

**Figure 12 sensors-22-02993-f012:**
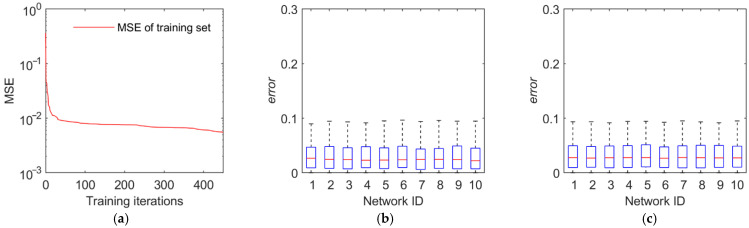
Training and test results of the identification probability. (**a**) Training convergence process of the best neural network. (**b**) Errors of the ten best networks on the training set. (**c**) Errors of the ten best networks on the test set.

**Figure 13 sensors-22-02993-f013:**
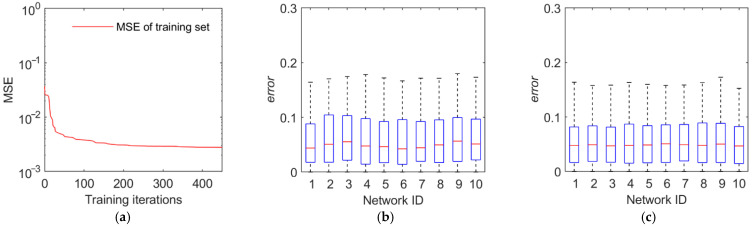
Training and test results of the tracking time percentage. (**a**) Training convergence process of the best neural network. (**b**) Errors of the ten best networks on the training set. (**c**) Errors of the ten best networks on the test set.

**Figure 14 sensors-22-02993-f014:**
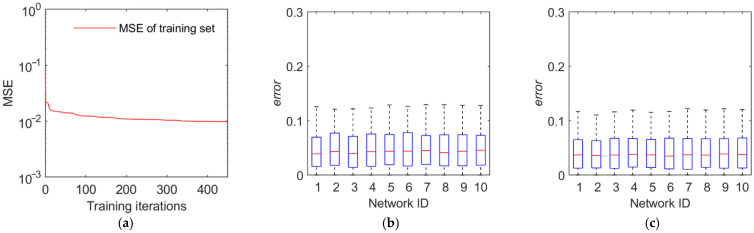
Training and test results of the average tracking interval. (**a**) Training convergence process of the best neural network. (**b**) Errors of the ten best networks on the training set. (**c**) Errors of the ten best networks on the test set.

**Figure 15 sensors-22-02993-f015:**
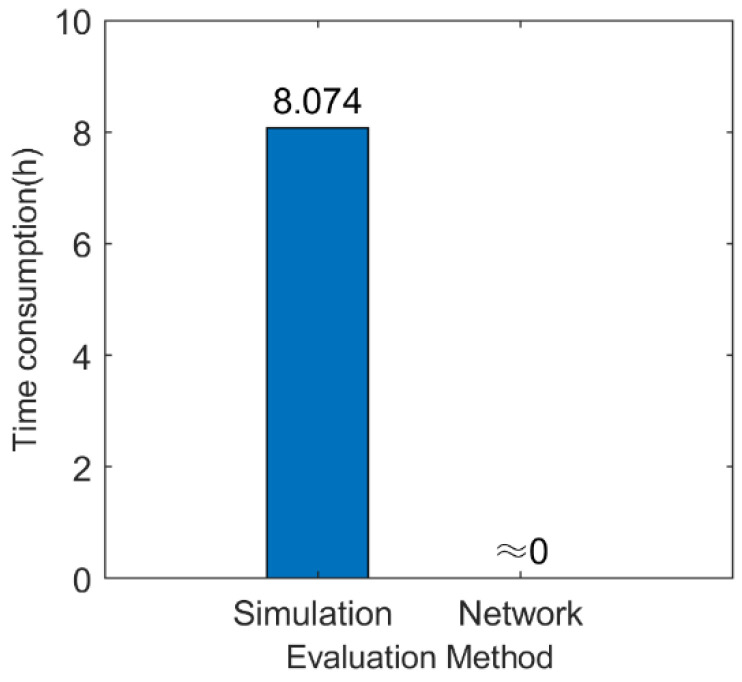
Time consumption of the simulation method and the neural network model.

**Table 1 sensors-22-02993-t001:** Parameters of the satellite cluster.

Category	Parameter	Value
Orbit parameter	Satellite number	100
	Orbital plane number	10
	Phase factor	1
	Orbit altitude (km)	650
	Eccentricity	0
	Inclination (°)	98
Attitude parameter	Maximum maneuver angle (°)	±45
	Maximum maneuver angular velocity (°/s)	1
	Maximum maneuver angular acceleration (°/s^2^)	0.1
	Attitude stabilization time (s)	10
Power parameter	Solar array output current (A)	15
	Average power in normal mode (W)	200
	Average power in attitude maneuver mode (W)	450
	Average power in imaging mode (W)	400
	Average power in data transmission mode (W)	270
Payload parameter	Payload type	Optical
	Imaging width of wide-swath satellites (km)	113.74
	Ground resolution of wide-swath satellites (m)	9.48
	Imaging width of high-resolution satellites (km)	22.69
	Ground resolution of high-resolution satellites (m)	1.89
	Payload data rate (Gbps)	22.69
	Solid-state memory capacity (GB)	500

**Table 2 sensors-22-02993-t002:** Environmental parameters for the model comparison.

Parameter	Symbol	Value Range
Simulation start time (day)	$t_{0}$	1
Longitude of initial area center (°)	Long	139.26
Latitude of initial area center (°)	Lat	16.44
Cloud level	$N_{c}$	4
Visibility level	$N_{v}$	4
Target number	$N_{Target}$	10

**Table 3 sensors-22-02993-t003:** The initial information of ten arbitrary ships.

Target ID	Position	Velocity (m/s)	Course Angle (°)
1	139.98° E, 17.83° N	3.34	36.49
2	139.53° E, 17.08° N	6.82	254.03
3	139.74° E, 16.33° N	6.79	160.27
4	140.25° E, 16.67° N	6.59	255.95
5	138.31° E, 17.63° N	1.97	314.51
6	140.00° E, 17.54° N	1.39	350.93
7	138.86° E, 17.62° N	8.42	44.62
8	139.84° E, 17.37° N	3.75	68.55
9	139.66° E, 17.40° N	3.88	182.86
10	139.32° E, 17.91° N	2.44	55.22

**Table 4 sensors-22-02993-t004:** Comparison of the three different simulation model granularities.

Granularity ID	Orbit	Attitude	Power	Data Transmission	Payload Coverage	Detection Probability
1	√	√			√	
2	√	√		√	√	Partially
3	√	√	√	√	√	√

**Table 5 sensors-22-02993-t005:** Evaluation results of the three different granularities.

Granularity ID	${\bar{P}}_{Discovery}$	${\bar{T}}_{Discovery}$ (s)	${\bar{P}}_{Identification}$	${\bar{T}}_{Identification}$ (s)	${\bar{A}}_{Tracking}$	${\bar{T}}_{Tracking}$ (s)
1	100.00%	2453.20	100.00%	574.90	16.89%	691.02
2	91.50%	2720.13	92.15%	1711.13	3.14%	2697.77
3	90.50%	2958.47	79.43%	2719.96	1.92%	3072.04

**Table 6 sensors-22-02993-t006:** The range of the sample input task parameters.

Parameter	Symbol	Value Range
Simulation start time (day)	$t_{0}$	[1, 365]
Longitude of initial area center (°)	Long	[130, 150]
Latitude of initial area center (°)	Lat	[10–30]
Cloud level	$N_{c}$	[1–6]
Visibility level	$N_{v}$	[1–6]
Target number	$N_{Target}$	[5–20]

**Table 7 sensors-22-02993-t007:** The input and output of the first ten samples.

Sample ID	Sample Input						Sample Output					
	$t_{0}$ (Day)	Long (°)	Lat (°)	$N_{c}$	$N_{v}$	$N_{Target}$	${\bar{P}}_{Discovery}$	${\bar{T}}_{Discovery}$ (s)	${\bar{P}}_{Identification}$	${\bar{T}}_{Identification}$ (s)	${\bar{A}}_{Tracking}$	${\bar{T}}_{Tracking}$ (s)
1	293	139.31	17.14	2	6	10	71.67%	4449.52	58.18%	4212.33	0.86%	3731.66
2	296	141.87	24.55	1	3	11	52.27%	6100.98	63.36%	2390.60	2.32%	2508.34
3	355	134.53	18.91	3	2	8	100.00%	5562.00	83.18%	1842.85	1.68%	3070.28
4	284	132.30	21.73	5	5	17	63.73%	6325.37	42.28%	3463.24	0.46%	4046.58
5	319	148.87	14.88	5	4	20	84.58%	3592.70	51.87%	3382.25	0.79%	4615.19
6	355	137.12	22.54	3	1	6	93.05%	5599.29	100.00%	1491.61	2.09%	3026.74
7	163	142.18	14.39	1	2	7	100.00%	5388.48	100.00%	777.75	2.54%	3027.65
8	141	137.75	20.09	4	6	15	28.89%	7854.62	43.04%	3196.58	0.65%	2904.51
9	128	145.41	19.13	1	4	10	95.00%	3522.58	81.77%	2095.00	1.69%	3697.10
10	237	139.67	17.64	6	3	12	0.00%	8956.60	57.84%	1831.08	1.02%	2522.32

**Table 8 sensors-22-02993-t008:** The training result of the activation function traversal.

Network ID	${\bar{P}}_{Discovery}$		${\bar{P}}_{Identification}$		${\bar{A}}_{Tracking}$		${\bar{T}}_{Tracking}$	
	Activation Function	Neuron Number	Activation Function	Neuron Number	Activation Function	Neuron Number	Activation Function	Neuron Number
1	softmax	40	softmax	20	softmax	80	softmax	20
2	elliotsig	20	softmax	40	softmax	40	tansig	40
3	satlin	40	softmax	80	softmax	100	softmax	60
4	poslin	20	softmax	100	softmax	60	logsig	20
5	softmax	80	logsig	20	satlins	20	satlins	20
6	purelin	80	logsig	40	softmax	20	logsig	60
7	purelin	40	softmax	60	logsig	40	softmax	40
8	purelin	60	satlin	20	purelin	20	softmax	100
9	logsig	40	tansig	60	purelin	40	elliotsig	20
10	purelin	20	satlins	20	purelin	60	poslin	20

**Table 9 sensors-22-02993-t009:** The best neural network of each effectiveness indicator.

Network	Hidden Layer Number	Activation Function	Neuron Number	MSE
${\bar{P}}_{Discovery}$	3	[softmax, softmax softmax]	[100, 100, 100]	0.0240
${\bar{P}}_{Identification}$	2	[softmax, softmax]	[160, 160]	0.0027
${\bar{A}}_{Tracking}$	4	[softmax, softmax, softmax, softmax]	[200, 200, 200, 200]	0.0091
${\bar{T}}_{Tracking}$	4	[softmax, softmax, softmax, softmax]	[60, 60, 60, 60]	0.0017

## Data Availability

The data used to support the findings of this study are available from the corresponding author upon request.

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
