# Peer review of "A Real-Time Effectiveness Evaluation Method for Remote Sensing Satellite Clusters on Moving Targets"

_sensors, 2022, doi:10.3390/s22082993_

Round 1

Reviewer 1 Report

The authors propose a real-time evaluation model architecture for remote sensing satellite clusters. The proposed approach includes a multi-physical field coupling simulation model of the satellite cluster,  a moving target observation indicator system, and a neural network for stakeholders to realize rapid evaluation. The presented experimental results show the effectiveness of the proposed method.

The manuscript contains some interesting research results. The presentation may be improved.

Comparison with other relevant methods is missing.

In figures 11, 12 and 14 the range of the values may be reduced in order to improve the quality of visualization.

Figure 15 may be omitted.

Author Response

Dear reviewer:

Thank you for your review report concerning our manuscript entitled “A real-time effectiveness evaluation method for remote sensing satellite clusters on moving targets” (ID: sensors-1646827). Those comments are all valuable and very helpful for revising and improving our paper. We have studied comments carefully and have made corrections which we hope meet with approval. The main corrections and responds to the comments are as following:

Point 1: Comparison with other relevant methods is missing.

Response 1: We are sorry that the comparison with other relevant methods is not obvious enough. However, in Section 6.1 and 6.4, this paper compares with other relevant studies. In Section 6.1, the accuracy and validity of the model proposed in this paper are proven by comparison with two mainstream simulation models, as shown in Figure 10. In section 6.4, by comparing the evaluation speed with the traditional method, the real-time performance of the evaluation model proposed in this paper is highlighted, which can be seen from Line 600-604.

Point 2: In figures 11, 12 and 14 the range of the values may be reduced in order to improve the quality of visualization.

Response 2: As reviewer suggested that, we slightly enlarged figure 11-14, and adjusted the ordinate range of B) and C) from [0, 1] to [0, 0.3].

Point 3: Figure 15 may be omitted.

Response 3: Considering the reviewer’s suggestion, we deleted Figure 15.

We appreciate for your warm work earnestly, and hope that the collection will meet with approvals.

Once again, thank you very much for your comments and suggestions.

Reviewer 2 Report

There are some specific comments, listed as follows.

  1. The text in Figures 4, 6 and 7 is blurry and difficult to read. Either improve the image quality or reduce the size of the image so that the text appears more clearly.
  2. Some of the symbols wrote with equation editor are larger in size, compared to letters in main text (like an example row 428, 439). Please, correct the size of the symbols.
  3. On Table 6, there are no spaces between Value Range – first and second row.

Author Response

Dear reviewer:

Thank you for your review report concerning our manuscript entitled “A real-time effectiveness evaluation method for remote sensing satellite clusters on moving targets” (ID: sensors-1646827). Those comments are all valuable and very helpful for revising and improving our paper. We have studied comments carefully and have made corrections which we hope meet with approval. The main corrections and responds to the comments are as following:

Point 1: The text in Figures 4, 6 and 7 is blurry and difficult to read. Either improve the image quality or reduce the size of the image so that the text appears more clearly.

Response 1: Considering the reviewer’s suggestion, we replaced figure 1-figure 7 with high-quality vector diagram to improve the clarity of the text.

Point 2: Some of the symbols wrote with equation editor are larger in size, compared to letters in main text (like an example row 428, 439). Please, correct the size of the symbols.

Response 2: In order to match the size of letters and symbols in the main text, we adjusted the font size of symbols written in the equation editor. Under the condition of meeting the format requirements of the journal template, we adjusted the font size of symbols in full-text from 10 points to 9 points.

Point 3: On Table 6, there are no spaces between Value Range – first and second row.

Response 3: We checked all the tables and supplemented the missing spaces in Table 2, 3, and 6.

We appreciate for your warm work earnestly, and hope that the collection will meet with approvals.

Once again, thank you very much for your comments and suggestions.

Reviewer 3 Report

The paper proposed a real-time evaluation model architecture for remote sensing satellite clusters. A multi-physical field coupling simulation model of the satellite cluster to observe moving targets is established firstly. Aside from considering the repercussions of on-board resource constraints, it also considers the consequences of the imaging's uncertainty effects on observation results. Secondly, a moving target observation indicator system is developed, which reflects the satellite cluster's actual effectiveness in orbit. Meanwhile, an indicator screening method using correlation analysis is proposed to improve the independence of the indicator system. Thirdly, a neural network is designed and trained for stakeholders to realize rapid evaluation. Different network structures and parameters are comprehensively studied to determine the optimized neural network model. Finally, based on the experiments carried out, the proposed neural network evaluation model can generate real-time, high-quality evaluation results.

The paper is well written, it presents the corresponding section. They are well addressed and written with detail. The text is well organized and easy to understand. The document presents examples on every step of the methodology and numbers when needed. Although I marked on the form that I don’t feel qualified to judge about the English language and style I found the document with a correct use of English. However, I let other reviewers to comment on this.

The conclusions are consistent with the work presented, it summarizes the methodologies, presents its results, and compares them with the methodologies already existent. Hence, I recommend it for publication after minor revision.

Author Response

Dear reviewer:

Thank you for your review report concerning our manuscript entitled “A real-time effectiveness evaluation method for remote sensing satellite clusters on moving targets” (ID: sensors-1646827). Those comments are all valuable and very helpful for revising and improving our paper. We have studied comments carefully and have made corrections which we hope meet with approval. The main corrections in the paper and the responds to the reviewer’s comments are summarized as following:

Point 1: The text in Figures 4, 6 and 7 is blurry and difficult to read. Either improve the image quality or reduce the size of the image so that the text appears more clearly.

Response 1: Considering the reviewer’s suggestion, we replaced figure 1-figure 7 with high-quality vector diagram to improve the clarity of the text.

Point 2: Some images (figure 9) should be brighter and on “figure 11-14 b) and c)” y scale should be changed.

Response 2: Considering the reviewer’s suggestion, we made the following corrections.

1) We enhanced Figure 8 and Figure 9 to make the picture brighter.

2) We slightly enlarged figure 11-14, and adjusted the ordinate range of B) and C) from [0, 1] to [0, 0.3].

Point 3: Figure 15 may be omitted.

Response 3: Considering the reviewer’s suggestion, we deleted Figure 15.

Point 4: Some of the symbols wrote with equation editor are larger in size, compared to letters in main text (like an example row 428, 439). Please, correct the size of the symbols.

Response 4: In order to match the size of letters and symbols in the main text, we adjusted the font size of symbols written in the equation editor. Under the condition of meeting the format requirements of the journal template, we adjusted the font size of symbols in full-text from 10 points to 9 points.

Point 5: On Table 6, there are no spaces between Value Range – first and second row.

Response 5: We checked all the tables and supplemented the missing spaces in Table 2, 3, and 6.

We appreciate for your warm work earnestly, and hope that the collection will meet with approvals.

Once again, thank you very much for your comments and suggestions.

Reviewer 4 Report

The main question addressed by the research is to evaluate effectiveness of remote sensing cluster on moving target and to develop real-time moving target observation indicator system.

The topic is relevant for this journal and readers because it shows development of multiple parts of simulation models, satellite cluster observation and neural network to observe moving target in real-time.

This article is first article, to the best of my knowledge, which is using this type of development for task of real-time moving target monitoring.

I have no further remarks regarding methodology and the conclusion is consistent with the evidence and presented arguments.

There is enough references and they are addressed properly and appropriate.

Tables are fine, only some images (figure 9) should be brighter and on “figure 11-14 b) and c)” y scale should be changed.

Author Response

Dear reviewer:

Thank you for your review report concerning our manuscript entitled “A real-time effectiveness evaluation method for remote sensing satellite clusters on moving targets” (ID: sensors-1646827). Those comments are all valuable and very helpful for revising and improving our paper. We have studied comments carefully and have made corrections which we hope meet with approval. The main corrections and responds to the comments are as following:

Point 1: Some images (figure 9) should be brighter and on “figure 11-14 b) and c)” y scale should be changed.

Response 1: Considering the reviewer’s suggestion, we made the following corrections.

1) We enhanced Figure 8 and Figure 9 to make the picture brighter.

2) We slightly enlarged figure 11-14, and adjusted the ordinate range of B) and C) from [0, 1] to [0, 0.3].

We appreciate for your warm work earnestly, and hope that the collection will meet with approvals.

Once again, thank you very much for your comments and suggestions.